# Effective purifying selection in ancient asexual oribatid mites

Alexander Brandt [1], Ina Schaefer[1], Julien Glanz[1], Tanja Schwander[2], Mark Maraun[1], Stefan Scheu[1,3] & Jens Bast [2]

Sex is beneficial in the long term because it can prevent mutational meltdown through increased effectiveness of selection. This idea is supported by empirical evidence of deleterious mutation accumulation in species with a recent transition to asexuality. Here, we study the effectiveness of purifying selection in oribatid mites which have lost sex millions of years ago and diversified into different families and species while reproducing asexually. We compare the accumulation of deleterious nonsynonymous and synonymous mutations between three asexual and three sexual lineages using transcriptome data. Contrasting studies of young asexual lineages, we find evidence for strong purifying selection that is more effective in asexual as compared to sexual oribatid mite lineages. Our results suggest that large populations likely sustain effective purifying selection and facilitate the escape of mutational meltdown in the absence of sex. Thus, sex per se is not a prerequisite for the long-term persistence of animal lineages.

[1] Johann-Friedrich-Blumenbach Institute of Zoology and Anthropology, Georg-August-University Goettingen, Untere Karspuele 2, DE-37073 Goettingen, Germany. [2] Department of Ecology and Evolution, University of Lausanne, UNIL Sorge, Le Biophore, CH-1015 Lausanne, Switzerland. [3] Center of Biodiversity and Sustainable Land Use, Georg-August-University Goettingen, Untere Karspuele 2, DE-37073 Goettingen, Germany. Correspondence and requests for materials should be addressed to A.B. (email: abrandt3@gwdg.de) or to J.B. (email: mail@jensbast.com)

One of the most challenging problems in evolutionary biology is to explain the maintenance of sex[1, 2]. Sex is coupled with substantial evolutionary costs, such as a twofold demographic cost due to the production of males, costs related to mate searching and mating (e.g. predator exposure and sexually transmitted diseases) and costs coupled with recombination (e.g. reduced likelihood of individual allele transmission and breakup of coadapted gene complexes)[3–5]. Despite these manifold costs, the mode of reproduction for the great majority of animal species is obligate sex. There is little agreement on which mechanisms generate selection for sex in natural populations in the short term[6]. However, theoretical predictions and empirical evidence for reduced purifying selection in asexual eukaryotes have led to the established consensus that sex and recombination are beneficial for the long-term persistence of lineages. This benefit derives from the ability of sexual reproduction to decouple linked loci with different fitness effects generated by genetic drift, which increases the effectiveness of purifying selection to purge mildly deleterious mutations and reduces Hill–Robertson effects[7–10]. Further, sex and recombination enable the restoration of the least loaded genotypes, which would otherwise be lost by chance (Muller's ratchet)[11].

Comparing rates of deleterious mutation accumulation in natural populations of sexual and asexual organisms is difficult as other factors than sex and recombination affect the effectiveness of selection. Most importantly, it is also determined by mutation rate and population size. Population size influences the strength of genetic drift in a population, which makes it an important determinant of mutational load and hence mutational decay of a population[12]. Nevertheless, empirical estimates showed increased accumulation of deleterious nonsynonymous mutations in effectively clonal lineages (hereafter referred to as asexual) as compared to closely related sexual lineages, e.g. in natural populations of *Timema* stick insects, *Campeloma* and *Potamopyrgus* freshwater snails and *Oenothera* evening primroses[13–17]. Nonsynonymous mutation accumulation occurred at up to 13.4 times the rate in sexual lineages[16]. Further, there is evidence for accumulation of nonsynonymous and synonymous mutations in non-recombining genomic regions of sexual organisms, such as mitochondria or (neo-) Y chromosomes[18, 19].

All asexual lineages that were analysed for mutation accumulation in previous studies lost sex relatively recently between 5000 years and 1.5 million years ago. Whether similar or even more prominent patterns are observed in long-term asexual populations, i.e. after tens of millions of years without sex, remains an open question. Few asexual lineages are known that have persisted and diversified in the absence of sex, most prominently bdelloid rotifers, darwinulid ostracods and various clades of oribatid mites[20, 21]. Two studies investigated the accumulation of deleterious nonsynonymous mutations in bdelloid rotifers[22, 23], but recent investigations indicate that bdelloid rotifers engage in non-canonical forms of sex and cannot be considered as effectively clonal[24–26]. Further, to our knowledge, there are no studies on mutation accumulation for darwinulid ostracods, only evidence of very slow background mutation rates[27]. Oribatid mites lost sex multiple times independently, several million years ago, followed by extensive radiation of parthenogenetic clades as indicated by their phylogenetic distribution and high inter- and intraspecific molecular divergence[28–32]. Thus this speciose, largely soil-living animal group (~10,000 species[20]) is well suited for comparative investigations of genomic consequences under long-term asexuality.

Here, we test if long-term asexuality in oribatid mites resulted in signatures of reduced effectiveness of selection by comparing the accumulation of deleterious nonsynonymous mutations and changes in codon usage bias (CUB). We analyse nuclear and mitochondrial orthologous genes from newly generated transcriptomic data of the three sexual species *Achipteria coleoptrata* (Linnaeus, 1758), *Hermannia gibba* (Koch, 1839) and *Steganacarus magnus* (Nicolet, 1855) and the three asexual species *Hypochthonius rufulus* (Koch, 1835), *Nothrus palustris* (Koch, 1839) and *Platynothrus peltifer* (Koch, 1839). We use two approaches to compare the effectiveness of purifying selection between asexual and sexual oribatid mite lineages on the transcriptome scale. First, we analyse the accumulation of nonsynonymous point mutations and infer their potential 'deleteriousness'. Second, we infer the effectiveness of selection on CUB since purifying selection also acts on synonymous sites[33]. Analyses are based on 3545 nuclear orthologous genes shared among the six species (see 'Methods'). Additionally, we analyse

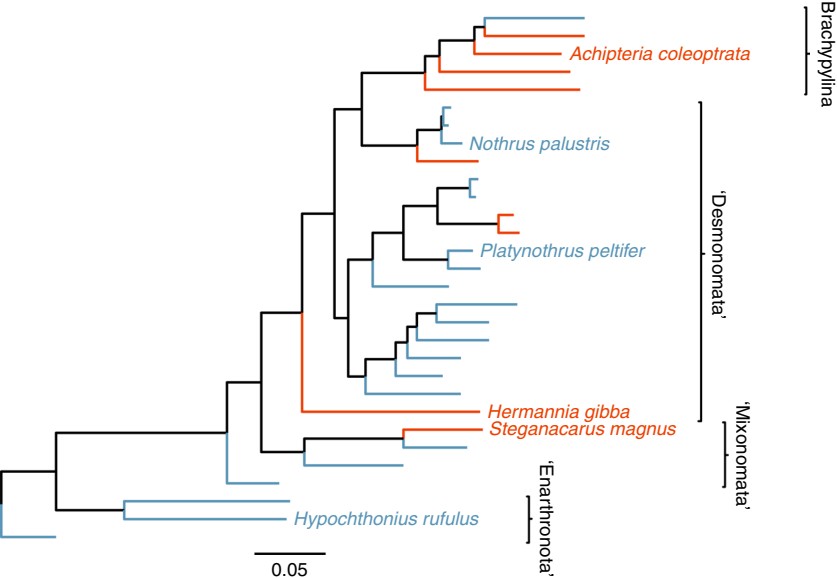

**Fig. 1** Phylogenetic tree of 30 oribatid mite species analysed in this study. The tree is based on partial sequences of 18S rDNA, *ef1α* and *hsp82* (see 'Methods'). The taxon sampling covers asexual (blue) and sexual (red) species from four out of six major groups of oribatid mites (brackets). Species used for transcriptome-wide analyses are indicated with full name, remaining species names are listed from top to bottom in Supplementary Table 1

two nuclear genes (*hsp82* and *ef1α*) of 30 species (nine sexual, 21 asexual species) from different phylogenetic groups[34] (Fig. 1). We find no evidence for reduced purifying selection in asexual lineages, neither for elevated accumulation of deleterious non-synonymous point mutations nor for 'relaxed' selection on CUB. Surprisingly, our results indicate even more effective purifying selection in asexual as compared to sexual oribatid mites.

## Results

**Accumulation of nonsynonymous mutations**. To estimate the accumulation rate of nonsynonymous point mutations in orthologs, we computed the ratio of nonsynonymous to synonymous divergence (dN/dS) as a measure of amino acid changes normalised for background mutation rates (see 'Methods'). Given the long divergence time between the six species used for sequencing of transcriptomes (>200 million years[35]), detectable orthologs shared among the six species are expected to be under strong purifying selection (dN/dS<1). In genes under strong purifying selection the majority of nonsynonymous changes are likely mildly deleterious, hence an increased accumulation of deleterious nonsynonymous mutations results in a higher dN/dS ratio[36]. We analysed nonsynonymous mutation accumulation exclusively at terminal branches of the phylogenetic tree, because character states (i.e. sexual or asexual reproduction) at internal branches are uncertain for methodological reasons[37]. We used a model of codon evolution estimating one dN/dS ratio for internal branches, one for terminal branches of asexual species and one for terminal branches of sexual species (from here referred to as terminal asexual and sexual branches, respectively; three-ratio model; see 'Methods'). As expected, all 3545 orthologs were under purifying selection at terminal branches (mean dN/dS = 0.084). Surprisingly and contrary to expectations, per-gene dN/dS ratios were on average lower at asexual branches as compared to sexual branches (mean $\Delta_{asex-sex} = -0.008$; Wilcoxon signed-rank test $V = 4.84 \times 10^{-6}$, $P < 0.001$; Fig. 2). Synonymous substitution rates (dS) did not differ between reproductive modes (means of 1.112 and 1.146 for asexual and sexual branches, respectively; gene effect $P = 0.951$, reproductive mode effect $P = 0.218$, interaction $P = 0.999$; permutation ANOVA), indicating that the differences in dN/dS ratios between reproductive modes were not driven by differences in dS.

Although orthologs overall were under stronger purifying selection in asexual as compared to sexual oribatid mites, dN/dS values as well as their difference between asexuals and sexuals varied widely among orthologs (ranges: dN/dS 0–0.627; $\Delta_{asex-sex}$ −0.172–0.540; variance: $4.6 \times 10^{-4}$). We therefore also analysed which specific orthologs differed significantly in their effectiveness of purifying selection between reproductive modes. To identify orthologs under significantly stronger purifying selection in asexual and sexual lineages, we tested if the three-ratio model was a better fit to the data than a model allowing for one dN/dS ratio for terminal and internal branches, without discriminating between reproductive modes (two-ratio model; see 'Methods'). For 67 orthologs, dN/dS ratios were significantly lower at asexual as compared to sexual branches and for five orthologs at sexual as compared to asexual branches (coloured bars; Fig. 2). The orthologs with strong selection in asexual mites were enriched for Gene Ontology terms related to e.g. immune response, mitotic cell cycle and germline cell division (Supplementary Data 1).

**Deleteriousness of nonsynonymous mutations**. In addition to elevated rates of nonsynonymous mutation accumulation, reduced purifying selection in sexual oribatid mite lineages is expected to result in fixation of nonsynonymous mutations that

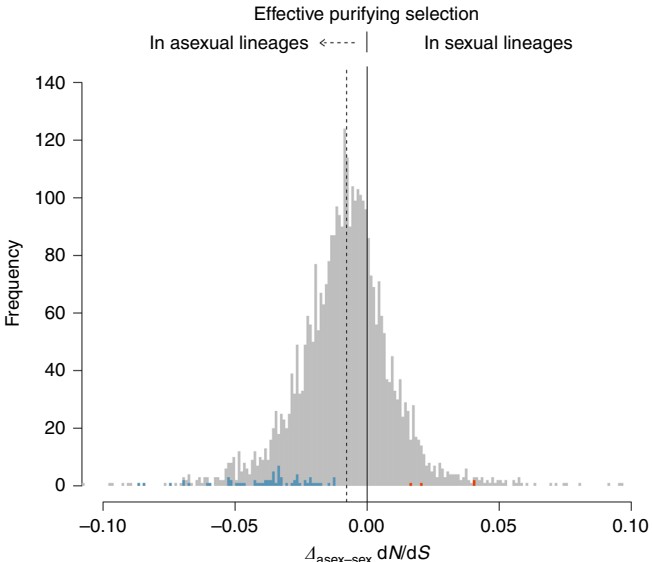

**Fig. 2** Per-gene differences in dN/dS ratios between reproductive modes. The histogram shows the distribution of per-gene differences in dN/dS ratios between asexual and sexual terminal branches in a tree comprising three sexual and three asexual oribatid mite species for 3539 orthologous loci under purifying selection (dN/dS < 1). To improve data presentation, the histogram range is limited to −0.1 to 0.1, representing 3539 out of 3545 orthologs. The distribution mean (−0.008; dotted line) is shifted to the left indicating overall lower dN/dS ratios at asexual as compared to sexual branches (Wilcoxon signed-rank test $V = 4.84 \times 10^6$, $P < 0.001$). Frequencies of genes with significantly lower dN/dS ratios at asexual as compared to sexual branches or sexual as compared to asexual branches are coloured blue and red, respectively

have stronger deleterious effects as compared to asexual lineages. We therefore analysed the potential 'deleteriousness' of the amino acid substitutions. For this, we inferred changes in hydrophobicity scores (HS) between ancestral and replacement amino acids as a measurement of the 'deleteriousness' of nonsynonymous mutations (see 'Methods'), because protein folding is influenced by changes in hydrophobic properties at amino acid substitution sites[38]. For nuclear orthologs, transitions between amino acids with more dissimilar hydrophobicity (HS < 90) involved 46,857 changes (50.74% of overall changes) at asexual and 68,115 (50.98%) at sexual branches. In accordance with the faster nonsynonymous mutation accumulation in sexual than asexual oribatid mites reported above, amino acids shifted to more dissimilar hydrophobicity at sexual branches (generalised linear mixed model $z = 2.4$, $P = 0.017$; Fig. 3a), indicating stronger 'deleteriousness' of nonsynonymous changes in sexual as compared to asexual oribatid mites.

**Analyses of the larger taxon sampling**. Overall, based on the analyses of transcriptomic orthologs shared among six species, asexual oribatid mite lineages accumulated less deleterious nonsynonymous mutations as compared to sexual lineages. However, the small number of taxa can generate branch length uncertainties which could affect inferences of dN/dS and ancestral amino acid sequences. To reduce this problem, we extended the taxon sampling to a total of 30 species. Using the same approaches, we compared nonsynonymous mutation accumulation among 21 asexual and nine sexual oribatid mite species (Fig. 1, Supplementary Table 1) in two nuclear genes (*ef1α* and *hsp82*) generated in a previous study[34]. Both genes were under strong purifying

selection at terminal branches as indicated by dN/dS ratios (0.045 and 0.034 for *ef1α* and *hsp82*, respectively). There were no significant differences between asexual and sexual branches as the two-ratio model provided a better fit to the data than the three-ratio model (Table 1). Synonymous substitution rates (dS) did not differ between reproductive modes, ($t = 0.627$, $P = 0.540$ and

$t = 1.246$, $P = 0.237$ for *ef1α* and *hsp82*, respectively; *t*-test). This lack of a significant difference between dN/dS ratios is unlikely the result of low variation in *ef1α* and *hsp82*, as mean pairwise divergence of amino acid sequences was high in both genes (0.069 and 0.109 for *ef1α* and *hsp82*, respectively). Consistent with dN/dS ratios, there were no differences in hydrophobicity changes at

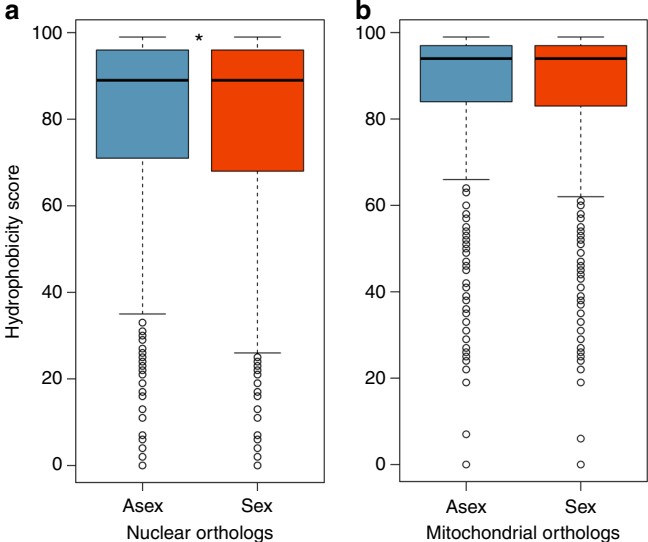

**Fig. 3** Hydrophobicity scores at asexual and sexual terminal branches. The boxplots show hydrophobicity scores (HS) at three asexual (blue) and three sexual (red) terminal branches for (**a**) 3545 nuclear orthologous genes (involving 92,351 and 133,606 HS at asexual and sexual terminal branches, respectively) and (**b**) ten mitochondrial orthologous genes (involving 1494 and 2468 HS at asexual and sexual terminal branches, respectively). HS measures the strength in changes of hydrophobicity from ancestral to replacement amino acids and indicates the 'deleteriousness' of a nonsynonymous mutation. The lower the HS the stronger is the change in hydrophobicity and 'deleteriousness' of a nonsynonymous mutation. For nuclear orthologs, amino acids shifted to more dissimilar hydrophobicity at sexual branches (generalised linear mixed model $z = 2.4$, $P = 0.017$), indicating stronger 'deleteriousness' of nonsynonymous changes in sexual as compared to asexual oribatid mites. For mitochondrial orthologs, there was no difference (generalised linear mixed model $z = 0.81$, $P = 0.421$). Significant differences in HS are marked with an asterisk (*$P < 0.05$). Whiskers correspond to 1.5 times the interquartile range

**Fig. 4** Comparison of CDC values between asexual and sexual oribatid mite species. Boxplots show CDC values of three asexual (blue) and three sexual (red) oribatid mite species for (**a**) 3545 nuclear orthologous genes (involving 10,635 CDC values for both asexual and sexual species) and (**b**) ten mitochondrial orthologous genes (involving 30 CDC values for both asexual and sexual species). CDC measures the deviation of observed from expected codon usage and allows for inference of the effectiveness of purifying selection acting on synonymous sites. A lower CDC value corresponds to more 'relaxed' selection on codon usage bias (see 'Methods'). For nuclear orthologs, per-gene CDC was slightly but significantly higher in asexual species (gene effect $P < 0.001$, reproductive mode effect $P = 0.008$, interaction $P = 0.616$; permutation ANOVA). For mitochondrial orthologs, per-gene CDC differed more strongly between reproductive modes (gene effect $P = 0.002$, reproductive mode effect $P < 0.001$, interaction $P = 0.960$; permutation ANOVA). Significant differences in CDC are marked with asterisks (***$P < 0.001$ and **$P < 0.01$). Whiskers correspond to 1.5 times the interquartile range

**Table 1 Results of the branch-specific dN/dS ratio analyses of *ef1α* and *hsp82* for 21 asexual and nine sexual oribatid mite species**

| Model | Free parameters | ln L | Likelihood ratio test | dN/dS |
|---|---|---|---|---|
| *ef1α* | | | | |
| 2 ratios (T/I) | 2 | −6115.434 | | T: 0.045<br>I: 0.036 |
| 3 ratios (sex_T/asex_T/I) | 3 | −6114.837 | vs. 2 ratios (T/I): $\chi^2 = 1.193$<br>$P = 0.275$ | sex_T: 0.051<br>asex_T: 0.041<br>I: 0.036 |
| *hsp82* | | | | |
| 2 ratios (T/I) | 2 | −4973.451 | | T: 0.034<br>I: 0.026 |
| 3 ratios (sex_T/asex_T/I) | 3 | −4972.167 | vs. 2 ratios (T/I): $\chi^2 = 2.567$<br>$P = 0.109$ | sex_T: 0.04<br>asex_T: 0.03<br>I: 0.026 |

asex_T terminal asexual branches, I internal branches, ln L ln(Likelihood), sex_T terminal sexual branches, T terminal branches
Non-significant P values indicate that the two-ratio model provides the better fit to the data than the three-ratio model for both genes. Therefore, asexual and sexual branches did not differ in the effectiveness of selection on nonsynonymous sites in these two genes

asexual and sexual branches for both genes ($W = 1637$, $P = 0.542$ and $W = 2235.5$, $P = 0.624$ for *ef1α* and *hsp82*, respectively; Wilcoxon rank-sum test).

As indicated by the transcriptome analyses described above, the dN/dS ratio differences between sexual and asexual taxa varied considerably among orthologs and only analyses of large gene sets are likely to reveal putative differences between reproductive modes. Indeed the gene *ef1α*, although present in the transcriptome ortholog set, did not feature significantly different dN/dS ratios between reproductive modes (0.015 and 0.026 at asexual and sexual branches, respectively; $\chi^2 = 3.157$, $P = 0.076$; likelihood ratio test). *Hsp82* was absent from the ortholog data set, preventing estimation of dN/dS ratios from the transcriptomes for this gene.

**Accumulation of synonymous mutations.** Although mutations at synonymous sites are generally considered to be neutral, they can be under selection for example because different codons influence the speed and accuracy of protein translation[33]. Hence, we also assessed the effectiveness of selection acting at synonymous sites by estimating CUB. This method is particularly robust because measurements of CUB do not depend on likelihood estimates, tree topologies or branch length estimates. As a metric, we used the codon deviation coefficient (CDC)[39], which calculates the deviation from expected CUB and accounts for background nucleotide composition, thus allowing for cross-species comparisons. A lower CDC value indicates more 'relaxed' selection on CUB (see 'Methods'). We compared gene-specific CDC values for each of the 3545 orthologous nuclear genes between reproductive modes. Consistent with the results for nuclear dN/dS ratios, per-gene CDC was slightly but significantly higher in asexual species (Fig. 4a; means of 0.130 and 0.128 for asexual and sexual species, respectively; gene effect $P < 0.001$, reproductive mode effect $P = 0.008$, interaction $P = 0.616$; permutation ANOVA), indicating more effective purifying selection at synonymous sites in asexuals.

**Accumulation of mutations in the mitochondria.** Next, we analysed mitochondrial genes, which allows for comparing the effectiveness of purifying selection among non-recombining genomes with different nuclear genomic backgrounds. We analysed the effectiveness of purifying selection in ten orthologous mitochondrial genes (*atp6*, *cob*, *cox1*, *cox2*, *cox3*, *nd1*, *nd2*, *nd3*, *nd4* and *nd5*) using the same approaches as for nuclear genes. Surprisingly, and contrasting to the results of nuclear orthologs, dN/dS ratios were lower at terminal sexual branches as compared to terminal asexual branches (mean $\Delta_{asex-sex} = 0.027$; Wilcoxon signed-rank test $V = 2$, $P = 0.006$). However, dS rates were highly elevated for sexual as compared to asexual branches (means of 2.814 and 5.577 for asexual and sexual branches, respectively; gene effect $P = 0.044$, reproductive mode effect $P = 0.002$, interaction $P = 0.876$; permutation ANOVA). This raises the question if the observed differences in mitochondrial dS between reproductive modes could be influenced by selection at synonymous sites[40]. Indeed, synonymous mitochondrial substitution rates (dS) and selection at synonymous sites as measured by CDC were negatively correlated in sexual lineages but not in asexual lineages ($r = -0.37$, $P = 0.042$ and $r = 0.16$, $P = 0.396$, respectively; Spearman's rank correlation). Because dS rates strongly differed and were adversely influenced by selection at synonymous sites between reproductive modes, mitochondrial dN/dS ratios are not comparable. Based on dN rates only, there was no difference in asexual and sexual nonsynonymous changes (means of 0.173 and 0.231 for asexual and sexual branches, respectively; gene effect $P = 0.037$, reproductive mode effect $P = 0.138$, interaction $P =$ 0.994; permutation ANOVA). Consistent with dN rates, the analysis of hydrophobicity changes ('deleteriousness') of amino acids for mitochondrial orthologs showed no overall difference between reproductive modes (generalised linear mixed model $z = 0.81$, $P = 0.421$; Fig. 3b). At mitochondrial synonymous sites, effectiveness of selection was strongly reduced for mitochondria in sexual lineages as compared to those in asexual lineages (Fig. 4b; mean CDC of 0.211 and 0.104 for asexual and sexual species, respectively; gene effect $P = 0.002$, reproductive mode effect $P < 0.001$, interaction $P = 0.960$; permutation ANOVA).

## Discussion

In metazoans it is believed that sex is important for the long-term persistence of a lineage because it facilitates effective purifying selection and rapid adaptation despite the effect of drift in finite populations. In contrast to this established consensus, our study showed no evidence for accumulation of deleterious nonsynonymous and synonymous point mutations in oribatid mite lineages that survived without sex for several million years. Moreover, our results suggest that purifying selection acts even more effectively in asexual as compared to sexual oribatid mite lineages. It is thus unlikely that asexual oribatid mite lineages drift towards extinction because of mutation accumulation. While our taxon sampling for dN/dS ratio estimates of nuclear orthologs in the transcriptome scale analyses was small, the overall pattern was substantiated by results from the more extended taxon set (Figs. 1 and 2, Table 1). Moreover, synonymous substitution analyses (CDC, Fig. 4) are independent of assumptions about phylogenetic relationships among taxa and these analyses likewise supported the result of more effective purifying selection in asexuals.

Our finding of more effective purifying selection despite the lack of sex is unique among investigated asexual eukaryotes. Previous studies that compared sexual and asexual eukaryotes typically found increased rates of nonsynonymous mutation accumulation in asexual as compared to sexual lineages, as predicted by theory (reviewed by Hartfield[17]). Where no differences were found, sex was lost probably too recently to detect accumulation of mutations[41, 42]. Increased rates of nonsynonymous mutation accumulation and 'relaxed' CUB further extends to self-fertilising eukaryotes, although selfers often maintain low rates of outcrossing[17, 43].

What mechanisms could account for the escape of asexual oribatid mites from the 'mutational meltdown'? Comparing the peculiarities of oribatid mites with those of animal groups including other old asexual lineages may help resolving this question. Oribatid mites are small animals (150–1400 μm) with a wide geographical distribution[44]. Both features also hold for other old asexual groups, such as darwinulid ostracods and meloidogyne root-knot nematodes, and contrast most asexual groups that accumulate deleterious mutations[45, 46]. Generally, small body size and large geographic range are indicative of high abundances of organisms[47, 48], which suggests that large population sizes might alleviate the negative effects of loss of sex.

Traditional models that predict mutation accumulation in asexuals are based on population sizes that are typical for many macroorganisms, but that might not be applicable to asexuals with large populations[7, 8, 11]. Indeed, large population sizes were proposed to maintain effective purifying selection in the absence of sex[49, 50], because the speed of mutation accumulation in asexual populations is predicted to drop substantially with increasing population sizes and eventually to become biologically irrelevant[20, 51, 52].

Consistent with the idea that large population size can protect asexual lineages from mutation accumulation, total oribatid mite densities are often very large and can reach up to ~350,000 ind. m$^{-2}$ in temperate and boreal forests. At high densities (>100,000 ind. m$^{-2}$) asexual species typically contribute >55% to total species numbers but >80% to total density[44], indicating substantially larger population sizes of asexual as compared to sexual oribatid mites. This observation is reflected by our findings of purifying selection being more effective in asexual oribatid mite lineages as compared to sexual lineages in both the nucleus and mitochondria. The observation of a strong difference for selection at synonymous sites for non-recombining mitochondria in sexual and asexual nuclear backgrounds supports the conclusion that asexual oribatid mite populations are larger than those of sexual oribatid mites (Fig. 4b). Remarkably, both animal population sizes and the frequency of parthenogenetic reproduction are generally high in soil animals (including protozoans, nematodes, enchytraeids, collembolans, isopods and oribatid mites[44, 53]), suggesting that more ancient asexual scandals may be hidden among these poorly studied groups.

Large population sizes can theoretically be sufficient to explain effective purifying selection in asexual oribatid mites, but additional non-mutually exclusive mechanisms could contribute to the reduction of the mutational load. Substitution rates can be altered via molecular mechanisms. For example, gene conversion has prevented mutation accumulation in the human Y chromosome and higher plant chloroplasts[54, 55] and DNA-repair contributes to the maintenance of DNA integrity in animal mitochondria[56]. However, as the CDC mainly reflects the impact of purifying selection rather than differences in mutation rates, large population size is likely the most important factor in preventing the accumulation of deleterious mutations in asexual oribatid mites. Finally, some non-canonical form of sex (e.g. horizontal gene transfer between bdelloid rotifers[26]) in oribatid mites cannot formally be excluded, although it seems unlikely that this would result in more effective selection than common bisexual reproduction.

In conclusion, we conducted several detailed analyses of mutation accumulation in transcriptomes of three asexual and sexual oribatid mite lineages and found purifying selection to be more effective in asexual lineages. This contrasts empirical evidence from younger asexual lineages and non-recombining genomic regions. Additional studies in other ancient asexual animal groups would allow testing of whether escaping deleterious mutation accumulation is a general feature of ancient asexuals. To what extent population sizes and/or molecular mechanisms contribute to the escape of mutational meltdown remains to be investigated. The results here indicate that deleterious mutation accumulation is unlikely to drive extant asexual oribatid mite lineages to extinction and challenge the established consensus that loss of sex necessarily leads to reduced effectiveness of purifying selection and mutational meltdown in the long term.

## Methods

**Animal sampling and sequencing of transcriptomes.** For analyses of transcriptomic data, we selected three sexual species (*A. coleoptrata*: Brachypylina, *H. gibba*: Desmonomata, *S. magnus*: Mixonomata) and three asexual species (*H. rufulus*: Enarthronota, *N. palustris*: Desmonomata, *P. peltifer*: Desmonomata) representing four of the six major oribatid mite groups[34] (Fig. 1). Animals were collected from litter in different forests in Germany (Goettinger Wald, Solling), extracted from litter alive using heat gradient extraction[57], identified following Weigmann[58] and starved for at least one week to avoid contamination by gut content RNA. For RNA extraction, between 1 and 50 individuals were pooled, depending on body size of the species, to obtain sufficient RNA for library construction while minimising the numbers of individuals pooled. These numbers were not biased towards sexual and asexual species. RNA extraction was done using Qiagen and ZymoResearch RNA extraction kits, following manufacturer's

instructions and RNA concentrations were measured using the Qubit 2.0 fluorometer (Invitrogen). Library preparation and sequencing of *A. coleoptrata*, *H. rufulus*, *N. palustris* and *H. gibba* was performed on Illumina HiSeq 2500 platforms, yielding 125 bp reads, by GATC Biotech (Constance, Germany) and of *P. peltifer* and *S. magnus* on Illumina MiSeq platforms, yielding 250 bp reads, by the Transcriptome and Genome Analysis Laboratory (TAL, Georg-August-University Goettingen, Germany). For detailed information, see Supplementary Table 2.

**Transcriptome assembly and data processing.** Quality of raw reads was controlled, checked for adaptors and the first 12 bases of each read were clipped and bases with quality scores below 30 were trimmed from the end of the reads using Trimmomatic version 0.33[59]. Trimmed read pairs were assembled with Trinity version 2.0.3[60] on the scientific computing clusters of the Georg-August-University Goettingen. The Trinity assembly process includes the reconstruction of genetic polymorphism resulting in different isoforms and is effective in avoiding the assembly of chimeric isoforms. Remains of adaptors and sequences matching to non-oribatid mite species in the NCBI database were removed. The cleaned assemblies were deposited under NCBI BioProject PRJNA339058. To avoid spurious contigs from sequencing errors, only contigs with RPKM (Reads Per Kilobase per Million mapped reads) > 2 were selected and the most abundant isoform per gene extracted following Harrison et al.[61]. The most abundant isoform represents the allele that is shared by most individuals in a pooled sample. As we analysed (long-term) fixed effects of mutation accumulation and not (short-term) segregating polymorphisms, the most abundant allele in the population is most suitable. To assess the quality of the generated transcriptomes and the subsequent RPKM filtered transcript set, we used BUSCO version 2.0[62] to score completeness and fragmentation of the assemblies (Supplementary Table 3). BUSCO uses a core set of genes that are present in single copy in 90% of arthropods for detection in the transcriptome set. All filtered assemblies contained a high fraction of full-length arthropod core genes ('complete'; >90%) and low levels of fragmentation, with the exception of *P. peltifer* (resulting from adaptor contamination in the reads that were trimmed). However, all mite assemblies were missing only very few core genes (<5%). Using the filtered assemblies, open reading frames were detected using TransDecoder version 2.0.1[63]. For subsequent analyses of branch-specific dN/dS ratios, CUB and hydrophobicity changes, orthologous ORFs were detected using reciprocal best alignment heuristics implemented in Proteinortho version 5.11[64] with amino acid sequences of detected ORFs as input. Amino acid sequences that were shared by all species were aligned with MUSCLE version 3.8.31[65] and back-translated using RevTrans version 1.4[66]. Alignments were curated (including deletion of triplet gaps) using Gblocks version 0.91b with parameters -t = c, -b = 4[67], alignments < 200 bp were excluded from further analyses, leaving 3545 alignments (mean alignment length 943 bp) for dN/dS ratio and codon usage analysis. As all analyses were based on alignments of orthologous ORFs from all six species, and curated to yield a uniform alignment size, the described differences in the assemblies did not affect selection estimates. For detailed information, see Supplementary Table 3. Mitochondrial genes were identified with blastn and tblastx and standard parameters using mitochondrial genes derived from genome data (Accession: PRJNA280488) and mitochondrial genes of *S. magnus* (Accession: EU935607) as query sequences[68, 69]. Alignment and alignment processing was done as described above for nuclear orthologous genes. For the general phylogeny and the analyses on the increased taxon sampling, we similarly aligned partial coding sequences of *ef1α*, *hsp82* and 18S rDNA of 30 oribatid mite species, generated in a previous study[34]. Sequences of these genes were available from NCBI, except for *N. palustris* for which sequences were extracted from the transcriptome data generated in this study using blast (Fig. 1, Supplementary Table 1).

**Phylogenetic tree estimations for dN/dS ratio analysis.** To reconstruct the phylogeny of the large taxon sampling of all 30 species (that includes the six species for which transcriptomes were generated), the alignments of *ef1α*, *hsp82* and 18S were concatenated and passed to RAxML version 3.1[70] with the GAMMAINVG model set as model of sequence evolution (Fig. 1). The non-coding 18S rDNA was included in a combined alignment of *ef1α* and *hsp82* for construction of the general phylogeny, as 18S rDNA is highly conserved among oribatid mites and improved the reliability of the resulting topology. This topology was used in subsequent analyses. For dN/dS ratio analyses, the fixed unrooted species tree, gene-specific alignments and gene-specific branch lengths are required. For nuclear and mitochondrial orthologs derived from the transcriptome data, the 30-taxa topology was compacted to contain the six species only and per-gene branch lengths were calculated using this fixed tree together with the processed alignments with RAxML and the GAMMAINVG model of sequence evolution. Similarly, for the large taxon set, branch lengths of *ef1α* and *hsp82* were calculated using the 30-species tree. For all gene-specific trees, branch lengths were multiplied with three to match the requirements of CodeML for testing models of codon evolution.

**Branch-specific dN/dS ratio analyses.** To infer nonsynonymous mutation accumulation in the three sexual and asexual lineages on transcriptome level, the alignments of 3545 orthologous genes were analysed using CodeML, implemented

in the PAML package version 4.8[71]. To get gene-specific dN, dS and dN/dS ratios, a custom script was used to pass each gene alignment together with the fixed species tree, appropriate branch labels (according to the model) and the respective individual branch lengths generated with RAxML to CodeML (available at https://github.com/ptranvan/mites2codeml). CodeML utilises a Maximum Likelihood framework to estimate the goodness of fit of a codon substitution model to the tree and the sequence alignment and calculates branch-specific dN/dS ratios. First, a model allowing for one dN/dS ratio for sexual, asexual and internal branches each was used to calculate dN/dS ratios (three-ratio model). The dN/dS ratios of terminal asexual and sexual branches were compared using a Wilcoxon signed-rank test in R version 3.0.2[72]. To investigate if the results were influenced by high synonymous substitution rates rather than by low nonsynonymous substitution rates, values of dS and dN were compared between terminal sexual and asexual branches using a non-parametric permutation ANOVA with 1000 bootstrap replicates, that does not require specific assumptions for distributions of data (as utilised in previous studies[16, 73]). To identify orthologous genes that were under significantly stronger purifying selection either at asexual or at sexual branches, the goodness of fit of the three-ratio model was compared to that of a model allowing for only one dN/dS ratio at terminal and internal branches, respectively (two-ratio model), using a likelihood ratio test (coloured bars, Fig. 2). Resulting P values were corrected for multiple testing using the R package qvalue[74]. To corroborate these analyses with a larger taxon set, branch-specific dN/dS ratios were predicted using the alignments of ef1α and hsp82 with the respective phylogenetic trees. Differences between terminal sexual and asexual branches were tested for significance as described above. To test if high synonymous mutation rates rather than low nonsynonymous mutation rates affected the results, values of dS and dN were compared between terminal sexual and asexual branches using a t-test. Additionally, mean pairwise amino acid divergences were calculated for ef1α and hsp82 to test if divergence was sufficiently high to detect possible differences between dN/dS ratios at terminal sexual and asexual branches.

**Analysis of mutation 'deleteriousness'.** To infer the 'deleteriousness' of nonsynonymous mutations, changes in hydrophobicity at amino acid replacement sites were analysed. To infer the ancestral states of amino acids for the nuclear orthologous genes, first Proteinortho was used to predict orthologous loci among translated ORFs of the six oribatid mite species and the transcriptome of Tetranychus urticae, that was used as outgroup (Accession: PRJNA315122)[75]. Orthologous ORFs were aligned using MUSCLE, back-translated using RevTrans and curated using Gblocks with similar parameters as above. Alignments < 200 bp were excluded from further analysis. Branch lengths were calculated for each gene, individually, using the processed alignments with fixed topologies including T. urticae as outgroup and the GAMMAINVG model of sequence evolution. For subsequent prediction of ancestral amino acids, branch lengths of individual trees were multiplied by three and the processed alignments were translated to amino acid using EMBOSS version 6.6.0[76]. A custom script was used to pass each translated alignment together with the fixed species tree and individual calculated branch lengths to CodeML with RateAncestor parameter set to 1 to predict ancestral amino acid sequences for each internal node (https://github.com/ptranvan/mites2codeml). Transitions from ancestral to replacement amino acids were scored for changes in hydrophobicity according to a hydrophobicity scoring matrix[77]. To infer the ancestral states of amino acids for mitochondrial genes, mitochondrial genes of T. urticae were downloaded from NCBI and added to the alignments as outgroup (Accession: NC_010526)[78]. Alignment and alignment processing was done as described above for nuclear orthologous genes. To infer the ancestral states of amino acids for the large taxon set, sequences of ef1α and hsp82 of the basal oribatid mite Palaeacarus hystricinus (Palaeosomata) were downloaded from NCBI (Accession: EF203793 and DQ090809, respectively) and added to the alignments as outgroup. Alignment and alignment processing was done as described above for nuclear orthologous genes. Hydrophobicity transitions were compared between modes of reproduction using generalised linear mixed models implemented in the R package lme4[79] with random effect of gene nested in species while correcting for overdispersion and fitted to Poisson distribution.

**Analysis of CUB.** Bias in codon usage was calculated with the CDC metric in Composition Analysis Toolkit version 1.3[39]. CDC estimates expected codon usage from observed positional GC and purine contents and calculates the deviation from observed codon usage using a cosine distance matrix, ranging from 0 (no deviation; no detectable (i.e. 'relaxed') selection on codon usage) to 1 (maximum deviation; effective selection on codon usage). To analyse CUB of nuclear and mitochondrial orthologous genes, we calculated CDC from processed alignments and compared per gene between reproductive modes using a permutation ANOVA similar to dS and dN comparisons (see above).

**Annotation and enrichment analysis.** To analyse enrichment of GO terms in genes that had significantly lower dN/dS ratios either at asexual as compared to sexual branches or at sexual as compared to asexual branches, the 3545 orthologous genes under purifying selection were annotated for the species H. rufulus using Blast2GO[80]. Automatic annotation was successful for 2587 genes, of which 67

genes yielded significantly lower dN/dS ratios at asexual as compared to sexual branches and five genes yielded significantly lower dN/dS ratios at sexual as compared to asexual branches. Based on the 2587 orthologous loci as reference set and the 67 and five genes as test sets, enrichment of GO terms was tested for the category Biological Process with Fisher's exact test in Blast2GO.

**Code availability.** The script for passing the input data to codeml and collecting the dN/dS estimate and hydrophobicity output is available under https://github.com/ptranvan/mites2codeml

**Data availability.** The transcriptomic data generated during this study are deposited in NCBI with the BioProject PRJNA339058. Further processed data that support the study are available upon request.

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

## Acknowledgements

We thank Christian Bluhm for help with species identification, Darren J. Parker for help with statistics, Patrick Tran Van for help with bioinformatics and Casper van der Kooi, Ken Kraaijeveld, Nicolas Galtier and Roy A. Norton for valuable comments on the manuscript. This study was supported by core funding of S.S., by DFG research fellowship BA 5800/1-1 to J.B. and Swiss SNF grant PP00P3_170627 to T.S.

## Author contributions

A.B. and J.B. conceived and designed the study. A.B., J.G. and J.B. collected samples. A.B. and J.G. performed wet lab work. A.B., J.G. and J.B. performed data analysis. T.S., I.S., M.M., S.S. contributed to data interpretation and analyses and A.B., J.B., I.S. wrote the paper with input from all authors.

## Additional information

**Competing interests:** The authors declare no competing financial interests.

