## [Peer Review File · Nature Communications]

Reviewers' comments:

Reviewer #1 (Remarks to the Author):

I read the manuscript of Brandt et al with great interest. The question why sex is the dominating reproductive mode in the eukaryotic world has kept evolutionary biologists busy for more than 20 years. The accumulation of slightly deleterious mutations is one of the expectations if sex and recombination have been absent for a long time, especially in combination with limited population sizes. With their novel genomic results, Brandt et al. thoroughly test this long standing hypothesis explaining the paradox of sex and, what is most puzzling, they actually find the opposite of what evolutionary theory predicts. The transcriptomic data of Brandt et al. from sexual and asexual oribatid mites clearly show that asexuals do not suffer from the accumulation of deleterious mutations ! That the exceptionally large population sizes of oribatid mites can explain this result, seems very reasonable to me.

There can be no doubt that this manuscript will not only attract a lot of interest from the evolutionary research community but also from a much wider scientific audience.

With its comparison between closely related sexuals and asexuals, the manuscript by Brandt et al. have an study excellent design at hand because other papers on the expected accumulation of deleterious mutations in asexuals have had more limited conclusions and lacked suitable comparisons between the reproductive modes or were limited to a small number of genes. The authors have used an impressive number of nuclear genes (more than 3500) to test the hypothesis of mutation accumulation in asexuals, and their conclusions are solid.

I have several smaller comments detailed below. Most of these are more technical and require additional details in the supplementary material, to increase the ability of other researchers for reproducing this work in similar biological systems.

My recommendation is "minor revision" and I am very much looking forward to seeing this manuscript published!

Minor comments:

Abstract, line 43: I would rather think that the accumulation of mutations will be more prominent in systems with ancient transitions to asexuality? I would therefore suggest to somehow rephrase this sentence.

Line 64: The costs are surely "evolutionary" costs?

Line 65: What is meant with "transmission" disadvantages and costs? This is not clear to non specialists.

Line 192-194: If there is indeed such a range of dN/dS ratios among orthologs, it will be necessary to somehow show the range and its distribution among orthologs. I would suggest an additional figure in the supplementary material showing the entire range and its frequency among the different orthologs.

Line 340: It will be necessary to show in more details, also in table 3 in the supplementary material, which reads were 125 and which 250 bp paired-end reads.

Line 408: I suggest to use another expression than "false discovery rates".

Figure 1 and general comment on the phylogenetic reconstructions: why were the 18S data

included for the tree construction? As these data were clearly not used for dN/dS ratio estimates, this requires at least some additional explanations in the material and method section.

Supplementary material:

Supplementary table 1: Do the authors mean that the EF1 α , hsp82 and 18S data of *Nothrus palustris* were included from the transcriptome data? If yes, then this should be explained more explicitly in the header of the table.

Supplementary table 2: I suggest to add the reproductive mode of the individual species to this table.

How did the authors deal with genetic variability among individuals given that up to 20 specimens were pooled prior to RNA extraction? I think that this point requires some additional details and attention because the procedure was different for the different species.

Likewise, transcriptomes were sequenced on two different Illumina platforms with different paired-end lengths. I think that details should be added on the possible effect of these different treatments on the quality and amount of obtained reads.

Supplementary table 3: My last point is reinforced from the data shown in this table for *Platynothrus peltifer*. Was the much lower N50 due to using the MiSeq, or shorter read lengths or both? How could the authors still obtain a comparable number of transcripts and filtered transcripts?

Gent, 5.4.17

Isa Schön

Reviewer #2 (Remarks to the Author):

The manuscript by Brandt et al. tests an important prediction in evolutionary biology related to the evolution of sex and recombination. Essentially it is predicted that deleterious mutations can be more efficiently removed in the presence of recombination, because such mutations can be shuffled onto different haplotypes. Thus, there could be evolutionary advantages to recombination. Here the authors test this prediction using oribatid mites. They examine several populations that have been asexual for a very long time as well as several sexual lineages. They find that the ratio of nonsynonymous to synonymous divergence appears to be slightly higher in the sexual lineages, suggested a greater burden of deleterious mutations in the sexual lineages. This is not what would have been predicted if recombination rate was the only factor leading to efficient removal of deleterious mutations. The authors offer several explanations for their findings including that the population sizes of the asexual lineages may be larger.

Overall, this is an interesting paper on an important topic. However, I have a number of substantial concerns about the logic of the study and several technical aspects of it too.

Major comments:

1. In the Introduction and Discussion, the authors never explain why sex and recombination are beneficial. Instead, they merely cite some work indicating this. The present manuscript would be much improved by better motivating the present work and why recombination could be evolutionarily advantageous. I would also suggest the authors cite several more relevant papers in this area, including:

Keightley and Otto, 2006 Nature
Felsenstein 1974 Genetics
Felsenstein and Yokoyama 1876 Genetics

2. Throughout the manuscript, the authors use the term "non-coding mutations" to refer to silent (or synonymous, or non amino-acid changing) mutations. This is a non-standard and improper definition of "non-coding". Synonymous changes are coding changes. Non-coding changes would be those outside of coding regions. The authors have not investigated such mutations here. Thus, all instances of "non-coding" must be changed to be "synonymous" or "silent".

3. The authors found that dN/dS was higher in the sexual lineages than in the asexual lineages. Could this pattern be explained by greater efficiency of positive selection in the sexual lineages? Theory predicts that recombination should allow more efficient positive selection. Positive selection can also increase dN/dS .

4. In the Discussion section, the authors suggest that the population sizes of the asexual lineages may be larger than those of the sexual lineages. This is a really important point, and in my view, could be the crux of the entire paper. I think the authors are trying to suggest that large population size can prevent the accumulation of deleterious alleles in the absence of recombination and allow the persistence of asexual lineages. This would be a very important conclusion. However, the authors need to explore this issue in greater depth. Simply adding it in the Discussion is not sufficient in my view. As an example of specific additional work, the authors could examine levels of neutral genetic variation in the sexual and asexual lineages. Are diversity levels higher in the asexual lineages, which would be consistent with there being a higher population size? Further, the manuscript should be re-written to bring up this issue of population size sooner.

5. Methods: How exactly did the authors arrive at the consensus sequence that was used for each species for the dN/dS analysis? It appears that between 1-50 individuals were pooled and sequenced. Presumably, there is some polymorphism among these many individuals. How was that dealt with? My biggest concern is that, when considering polymorphisms, dN/dS is highly sensitive to sample size. In larger samples, dN/dS will be higher than in smaller samples. This occurs because nonsynonymous changes are more deleterious and tend to be at lower frequency than synonymous changes. As such, by

increasing the sample size, a greater proportion of the rare nonsynonymous changes are detected, leading to an apparent increase in dN/dS . I worry about this issue in this particular study. If more individuals were sampled from the sexual lineages, then that could lead to dN/dS appearing to be higher in the sexual lineages than the asexual lineages.

Minor comments:

1) Page 3, line 80: Human Y-chromosomes actually have much lower diversity than predicted by standard neutral models or models of complex demography. See Wilson Sayres et al. (2014) PLoS Genetics.

2) Page 5, line 126-127: The authors write that "any coding change is likely deleterious." This is likely overly simplistic. A gene can be under purifying selection, however, it is not required that mutations at every site would be deleterious. This sentence should be revised.

3) Top of page 7: The authors write that "reduced purifying selection is expected to translate into coding mutations with strong deleterious effects". The way that this sentence is worded is technically incorrect. Reduced purifying selection does not result in coding mutations. Assuming that the mutation rate is not changed (which appears to be the assumption throughout this study), then the rate of appearance of deleterious mutations is the same. Reduced purifying selection affects what happens to those deleterious mutations in the population. Reduced purifying selection can increase the fixation probability of the deleterious mutations, allowing them to accumulate as substitutions. Also, I don't think the authors can conclude that the mutations (or substitutions) would have "strong deleterious effects". Relaxed purifying selection is more likely to allow weakly deleterious mutations to persist. Strongly deleterious ones will be purged out, regardless of the demography of the population. This sentence should be revised in light of my comments.

Reviewer #1 (Remarks to the Author):

I read the manuscript of Brandt et al with great interest. The question why sex is the dominating reproductive mode in the eukaryotic world has kept evolutionary biologists busy for more than 20 years. The accumulation of slightly deleterious mutations is one of the expectations if sex and recombination have been absent for a long time, especially in combination with limited population sizes. With their novel genomic results, Brandt et al. thoroughly test this long standing hypothesis explaining the paradox of sex and, what is most puzzling, they actually find the opposite of what evolutionary theory predicts. The transcriptomic data of Brandt et al. from sexual and asexual oribatid mites clearly show that asexuals do not suffer from the accumulation of deleterious mutations ! That the exceptionally large population sizes of oribatid mites can explain this result, seems very reasonable to me.

There can be no doubt that this manuscript will not only attract a lot of interest from the evolutionary research community but also from a much wider scientific audience.

With its comparison between closely related sexuals and asexuals, the manuscript by Brandt et al. have an study excellent design at hand because other papers on the expected accumulation of deleterious mutations in asexuals have had more limited conclusions and lacked suitable comparisons between the reproductive modes or were limited to a small number of genes. The authors have used an impressive number of nuclear genes (more than 3500) to test the hypothesis of mutation accumulation in asexuals, and their conclusions are solid.

I have several smaller comments detailed below. Most of these are more technical and require additional details in the supplementary material, to increase the ability of other researchers for reproducing this work in similar biological systems.

My recommendation is "minor revision" and I am very much looking forward to seeing this manuscript published!

We very much appreciate the positive evaluation of our manuscript.

Minor comments:

1. Abstract, line 43: I would rather think that the accumulation of mutations will be more prominent in systems with ancient transitions to asexuality? I would therefore suggest to somehow rephrase this sentence.

We agree that according to theory deleterious mutation accumulation will be more prominent in old asexual lineages. The sentence aimed at emphasising that empirical evidence for deleterious mutation accumulation is restricted to species with a relatively recent transition to asexuality (< 1.5 my). This highlights the need for investigating a system with an ancient transition to asexuality. Given the word limitation we left the abstract as is and refer to lines 94-96 in the introduction, where we clarified this point: “Whether similar or even more prominent patterns are observed in long-term asexual populations, i.e. after tens of millions of years without sex remains an open question.”

2. Line 64: The costs are surely "evolutionary" costs?

We reformulated the sentence according to the referee’s suggestion (lines 64-68). It reads now: “Sex is coupled with substantial evolutionary costs, such as a twofold demographic cost due to the production of males, costs related to mate searching and mating (e. g. predator exposure and sexually transmitted diseases) and costs coupled with recombination (e. g. reduced likelihood of individual allele transmission and breakup of coadapted gene complexes)³⁻⁵.”

3. Line 65: What is meant with "transmission" disadvantages and costs? This is not clear to non specialists.

We changed the sentence to explain costs related to sexual reproduction in more detail (lines 64-68: see comment above).

4. Line 192-194: If there is indeed such a range of dN/dS ratios among orthologs, it will be necessary to somehow show the range and its distribution among orthologs. I would suggest an additional figure in the supplementary material showing the entire range and its frequency among the different orthologs.

We thank the referee for this suggestion. We limited the range of the x-axis in figure 2 to -0.1 to 0.1 for better representing the ortholog variance in the figure as only six values out of 3,545 of $\Delta_{\text{asex-sex}}$ fell outside of the range (i.e. -0.172; -0.108; 0.111; 0.205; 0.256; 0.540). We added this information to the legend of figure 2 (lines 542, 543). It reads now: “To improve data presentation, the histogram range is limited to -0.1 to 0.1, representing 3,539 out of 3,545 orthologs.”

We also added the total range and variance to the results (lines 153-156): “Although orthologs overall were under stronger purifying selection in asexual as compared to sexual oribatid mites, dN/dS values as well as their difference between asexuals and sexuals varied widely among orthologs (ranges: dN/dS 0-0.627; $\Delta_{\text{asex-sex}}$ -0.172-0.540; variance: 0.00046).”

5. Line 340: It will be necessary to show in more details, also in table 3 in the supplementary material, which reads were 125 and which 250 bp paired-end reads.

We added information on sequencing platform and read length used for the respective species to the sentence (lines 347-352): “Library preparation and sequencing of *A. coleoptrata*, *H. rufulus*, *N. palustris* and *H. gibba* was performed on Illumina HiSeq 2500 platforms, yielding 125 bp reads, by GATC Biotech (Constance, Germany) and of *P. peltifer* and *S. magnus* on Illumina MiSeq platforms, yielding 250 bp reads, by the Transcriptome and Genome Analysis Laboratory (TAL, Georg-August-University Göttingen, Germany).”

Further, we added the requested information on read length to Supplementary Table 2.

6. Line 408: I suggest to use another expression than "false discovery rates".

We changed the expression “false discovery rate adjusted” to “corrected for multiple testing” (line 439).

7. Figure 1 and general comment on the phylogenetic reconstructions: why were the 18S data included for the tree construction? As these data were clearly not used for dN/dS ratio estimates, this requires at least some additional explanations in the material and method section.

As 18S rDNA is highly conserved among oribatid mite species, we combined the ef1 α , hsp82 and 18S rDNA alignments to increase the reliability of the general phylogeny. The resulting topology served as fixed species tree used for dN/dS estimates of ef1 α , hsp82 and the 3,545 orthologs with gene-specific branch-lengths. We added more details on this procedure to the Methods section (lines 405-408): “The non-coding 18S rDNA was included in a combined alignment of ef1 α and hsp82 for construction of the general phylogeny, as 18S rDNA is highly conserved among oribatid mites and improved the reliability of the resulting topology. This topology was used in subsequent analyses.”

8. Supplementary table 1: Do the authors mean that the EF1 α , hsp82 and 18S data of *Nothrus palustris* were included from the transcriptome data? If yes, then this should be explained more explicitly in the header of the table.

Yes. We added additional information on the sequences of Nothrus palustris that were extracted from the transcriptome data to the header of Supplementary Table 1 (lines 11-13): “Because sequences of *Nothrus palustris* were not part of the study of Domes et al. (2007) and not available from NCBI, they were extracted from the transcriptome data.”

9. Supplementary table 2: I suggest to add the reproductive mode of the individual species to this table. How did the authors deal with genetic variability among individuals given that up to 20 specimens were pooled prior to RNA extraction? I think that this point requires some additional details and attention because the procedure was different for the different species.

Likewise, transcriptomes were sequenced on two different Illumina platforms with different paired-end lengths. I think that details should be added on the possible effect of these different treatments on the quality and amount of obtained reads.

The Trinity assembly process includes the reconstruction of genetic polymorphism resulting in different isoforms and is effective in minimising the risk of assembling chimeric isoforms. Using an RPKM based approach, we identified the most abundant isoform per gene representing the most abundant allele in a pooled sample. As we elucidate (long-term) fixed effects of mutation accumulation, and not (short-term) segregating polymorphisms, the most abundant allele in the population is most suitable for these analyses. We added this information to the methods at lines 360-362 and 367-370. It reads now: “The Trinity assembly process includes the reconstruction of genetic polymorphism resulting in different isoforms and is effective in avoiding the assembly of chimeric isoforms.” and “The most abundant isoform represents the allele that is shared by most individuals in a pooled sample. As we analysed (long-term) fixed effects of mutation accumulation and not (short-term) segregating polymorphisms, the most abundant allele in the population is most suitable.”

*The numbers of individuals pooled for RNA extraction were not biased towards reproductive mode (for asexual species 50, 3 and 1, for sexual species 20, 10 and 1 individuals). We added information on the reproductive mode to Supplementary Table 2. Further, sequences of *ef1a* and *hsp82* used for the analyses of the 30 taxa sampling were from single individuals and supported the analyses of transcriptomic orthologs. Analyses of selection at synonymous sites that segregate at higher frequencies than nonsynonymous sites (CDC data, Fig. 4) likewise supported the finding of more effective selection in asexuals as compared to sexuals.*

*To assess the quality of the generated transcriptomes and the subsequent RPKM filtered transcript set, we added BUSCO analyses to the study to score the completeness and fragmentation of the assemblies. BUSCO uses a core set of genes that are present in single copy in 90 % of arthropods for detection in the transcriptome set. All filtered assemblies were found to contain a high fraction of arthropod core genes assembled to full length (‘complete’; > 90 %) and low levels of fragmentation, i. e. genes split into several transcripts (< 5.4 %), with the exception of *P. peltifer* (83.4 % and 11.7 %). However, all mite assemblies missed only few core genes (< 5 %). According to the consistency of these results, the variable number of individuals and different sequencing platforms (added to Supplementary Table 2) with a variable number of reads and read lengths (added to Supplementary Table 3) did not influence the assembly quality, as metrics did not differ between *Steganacarus magnus* (MiSeq) and *Achipteria coleoptrata*, *Hermannia gibba*, *Hypochothonius rufulus* and *Nothrus palustris* (HiSeq 2500). The reduced N50 and more fragmented state of the *Platynothrus peltifer* assembly likely originated from read contamination with adaptors that were removed prior to the assembly. However, the reduced assembly quality of *P. peltifer* did not result in more missing genes (BUSCO ‘missing’ genes are within the range of all assemblies < 5.1 %). We added details on the analyses with BUSCO to the manuscript (lines 370-378): “To assess the quality of the generated transcriptomes and the subsequent RPKM filtered transcript set, we used BUSCO version 2.0⁶⁹ to score completeness and fragmentation of the assemblies (see Supplementary Table 2). BUSCO uses a core set of genes that are present in single copy in 90% of arthropods for detection in the transcriptome set. All filtered assemblies contained a high fraction of full length arthropod core genes (‘complete’; > 90%) and low levels of fragmentation, with the exception of *P. peltifer* (resulting from adaptor contamination in the reads that were trimmed). However, all mite assemblies were missing only few core genes (< 5%).”*

Further, we added the results of the BUSCO analyses to Supplementary Table 3 and indicated this in the header (lines 261,262): “The filtered transcript set was assessed for quality using BUSCO (‘complete’ are core arthropod genes assembled to full length).”

Moreover, selection estimates are based on alignments of orthologous ORFs from all six species followed by gap trimming, yielding a uniform alignment size. Thus, for all species, the same region of the ortholog was used, and consequently the differences in assembly did not affect selection estimates. We now included this at lines 387-389: “As all analyses were based on alignments of orthologous ORFs from all six species, and curated to yield a uniform alignment size, the described differences in the assemblies did not affect selection estimates.”

10. Supplementary table 3: My last point is reinforced from the data shown in this table for *Platynothrus peltifer*. Was the much lower N50 due to using the MiSeq, or shorter read lengths or both? How could the authors still obtain a comparable number of transcripts and filtered transcripts?

The reduced N50 of Platynothrus peltifer likely resulted from read contamination with adaptors that were removed prior to the assembly (see comment above). As the P. peltifer assembly did not miss more genes compared to the other species assemblies, and contained only 5 % more transcripts assembled in fragments, the number of identified ORFs was similar to the other assemblies.

Reviewer #2 (Remarks to the Author):

The manuscript by Brandt et al. tests an important prediction in evolutionary biology related to the evolution of sex and recombination. Essentially it is predicted that deleterious mutations can be more efficiently removed in the presence of recombination, because such mutations can be shuffled onto different haplotypes. Thus, there could be evolutionary advantages to recombination. Here the authors test this prediction using oribatid mites. They examine several populations that have been asexual for a very long time as well as several sexual lineages. They find that the ratio of nonsynonymous to synonymous divergence appears to be slightly higher in the sexual lineages, suggested a greater burden of deleterious mutations in the sexual lineages. This is not what would have been predicted if recombination rate was the only factor leading to efficient removal of deleterious mutations. The authors offer several explanations for their findings including that the population sizes of the asexual lineages may be larger.

Overall, this is an interesting paper on an important topic. However, I have a number of substantial concerns about the logic of the study and several technical aspects of it too.

We very much appreciate the positive evaluation of our manuscript.

Major comments:

1. In the Introduction and Discussion, the authors never explain why sex and recombination are beneficial. Instead, they merely cite some work indicating this. The present manuscript would be much improved by better motivating the present work and why recombination could be evolutionarily advantageous. I would also suggest the authors cite several more relevant papers in this area, including:

Keightley and Otto, 2006 Nature

Felsenstein 1974 Genetics

Felsenstein and Yokoyama 1876 Genetics

We thank the referee for these suggestions. We have added details and the requested references on the advantage of sexual reproduction to the introduction at lines 73-78: “This benefit derives from the ability of sexual reproduction to decouple linked loci with different fitness effects generated by genetic drift, which increases the effectiveness of background selection to purge mildly deleterious mutations (Hill-Robertson effect)⁸⁻¹¹. Further, sex and recombination allow to restore least loaded genotypes that would otherwise be lost by chance (Muller’s ratchet)¹².”

To further improve the motivation of the study we stress that the scientific consensus on the long-term advantages of sexual reproduction is based on estimates of mutation accumulation in asexuals with a recent transition to asexuality but lack evidence from old asexual lineages. We emphasise this at lines 94-96 and 108-110: “Whether similar or even more prominent patterns are observed in long-term asexual populations, i. e. after tens of millions of years without sex remains an open question.” and “Here, we tested if long-term asexuality in oribatid mites resulted in signatures of reduced effectiveness of selection by comparing the accumulation of deleterious nonsynonymous mutations and changes in codon usage bias.”

2. Throughout the manuscript, the authors use the term “non-coding mutations” to refer to silent (or synonymous, or non amino-acid changing) mutations. This is a non-standard and improper definition of “non-coding”. Synonymous changes are coding changes. Non-coding changes would be those outside of coding regions. The authors have not investigated such mutations here. Thus, all instances of “non-coding” must be changed to be “synonymous” or “silent”.

We changed ‘non-coding’ to ‘synonymous’ and ‘coding’ to ‘nonsynonymous’ throughout the manuscript.

3. The authors found that dN/dS was higher in the sexual lineages than in the asexual lineages. Could this pattern be explained by greater efficiency of positive selection in the sexual lineages? Theory predicts that recombination should allow more efficient positive selection. Positive selection can also increase dN/dS.

The orthologs used for the analyses in this study are shared among all six species, which diverged more than 200 million years ago. Thus, as they were detectable in the transcriptomes, they must be conserved and under strong purifying selection. This is reflected by all 3,545 orthologs yielding dN/dS ratios << 1 and low average dN/dS for sexuals (0.088) and asexuals (0.079).

In such highly conserved genes, diversifying selection could still act on individual sites. However, we chose a method (CodeML) that estimates dN/dS of entire genes, such that positive site-specific selection is not reflected in the dN/dS values.

To formally address the influence of positive selection, we used the program BUSTED (Murrell et al. 2015; part of the HyPhy package) to detect positive selection that has acted at a subset of sites on any of the terminal branches (regardless of sexual or asexual taxa) in the 3,545 orthologs. We found a maximum of 300 orthologs with at least one branch being likely under site-specific episodic diversification. Removing this small number of orthologs from the analyses yielded similar results (mean $\Delta dN/dS_{\text{asex-sex}} = -0.0084$ vs. -0.0081 of the full set).

In conclusion, these points show that positive selection does not influence the results.

4. In the Discussion section, the authors suggest that the population sizes of the asexual lineages may be larger than those of the sexual lineages. This is a really important point, and in my view, could be the crux of the entire paper. I think the authors are trying to suggest that large population size can prevent the accumulation of deleterious alleles in the absence of recombination and allow the persistence of asexual lineages. This would be a very important conclusion. However, the authors need to explore this issue in greater depth. Simply adding it in the Discussion is not sufficient in my view. As an example of specific additional work, the authors could examine levels of neutral genetic variation in the sexual and asexual lineages. Are diversity levels higher in the asexual lineages, which would be consistent with there being a higher population size? Further, the manuscript should be re-written to bring up this issue of population size sooner.

We thank the referee for these suggestions. We added additional explanation on the impact of population sizes to the introduction (lines 82-84). It reads now: "Population size influences the strength of genetic drift in a population, which makes it an important determinant for the speed of mutational decay through the Hill-Robertson effect and Muller's ratchet in asexual lineages¹³⁻¹⁵."

Indeed, it would be very interesting and important to follow up on this study with specific tests for the effect of population sizes on the effectiveness of selection in asexual mite populations. Unfortunately, the design of this study does not allow testing for differences in neutral genetic variation, as we pooled different numbers of individuals. Our study focused on investigating long-term effects of asexual reproduction that are reflected by the accumulation of fixed mutations, so we did not include comparisons of polymorphisms.

Given the surprising results of more effective purifying selection in asexuals as compared to sexuals, which contrasts all previous studies, we provide potential explanations for this pattern. We do not claim to have proof for any of the three discussed explanations (large population sizes, molecular mechanisms and/or cryptic sex) but large population sizes might be a major factor as discussed in the manuscript. Earlier studies indicated that overall population densities of asexual species exceed those of sexual species (lines 297-300): "At high densities ($> 100,000 \text{ ind/m}^2$) asexual species typically contribute $> 55\%$ to total species number but $> 80\%$ to total density⁵⁰, indicating substantially larger population sizes of asexual as compared to sexual oribatid mites."

Further, selection on synonymous sites (CDC data) of non-recombining mitochondria of asexual species in fact strongly exceeds those of sexual species (Fig. 4b). As we cannot test for differences

of population sizes and effectiveness of selection per se, these explanations are only considered in the Discussion and not the Results section.

5. Methods: How exactly did the authors arrive at the consensus sequence that was used for each species for the dN/dS analysis? It appears that between 1-50 individuals were pooled and sequenced. Presumably, there is some polymorphism among these many individuals. How was that dealt with? My biggest concern is that, when considering polymorphisms, dN/dS is highly sensitive to sample size. In larger samples, dN/dS will be higher than in smaller samples. This occurs because nonsynonymous changes are more deleterious and tend to be at lower frequency than synonymous changes. As such, by increasing the sample size, a greater proportion of the rare nonsynonymous changes are detected, leading to an apparent increase in dN/dS. I worry about this issue in this particular study. If more individuals were sampled from the sexual lineages, then that could lead to dN/dS appearing to be higher in the sexual lineages than the asexual lineages.

We refer to question 9 of referee 1 who raised similar concerns (copied & pasted here for convenience). We think that an influence of sample size on dN/dS is highly unlikely in this particular study for the following reasons:

The Trinity assembly process includes the reconstruction of genetic polymorphism resulting in different isoforms and is effective in minimising the risk of assembling chimeric isoforms. Using an RPKM based approach, we identified the most abundant isoform per gene representing the most abundant allele in a pooled sample. As we elucidate (long-term) fixed effects of mutation accumulation, and not (short-term) segregating polymorphisms, the most abundant allele in the population is most suitable for these analyses. We added this information to the methods at lines 360-362 and 367-370. It reads now: “The Trinity assembly process includes the reconstruction of genetic polymorphism resulting in different isoforms and is effective in avoiding the assembly of chimeric isoforms.” and “The most abundant isoform represents the allele that is shared by most individuals in a pooled sample. As we analysed (long-term) fixed effects of mutation accumulation and not (short-term) segregating polymorphisms, the most abundant allele in the population is most suitable.”

The numbers of individuals pooled for RNA extraction were not biased towards reproductive mode (for asexual species 50, 3 and 1, for sexual species 20, 10 and 1 individuals). We added information on the reproductive mode to Supplementary Table 2. Further, sequences of *ef1a* and *hsp82* used for the analyses of the 30 taxa sampling were from single individuals and support the analyses of transcriptomic orthologs. Analyses of selection at synonymous sites that segregate at higher frequencies than nonsynonymous sites (CDC data, Fig. 4) likewise support the finding of more effective selection in asexuals as compared to sexuals.

Minor comments:

1) Page 3, line 80: Human Y-chromosomes actually have much lower diversity than predicted by standard neutral models or models of complex demography. See Wilson Sayres et al. (2014) PLoS Genetics.

We clarified that the empirical evidence we refer to derives from the comparison of neo-Y with neo-X chromosomes (in Drosophila miranda; lines 89-92). It reads now: “Further, there is evidence for accumulation of nonsynonymous and synonymous mutations in non-recombining genomic regions of sexual organisms, such as mitochondria or (neo-) Y chromosomes^{21,22}.”

2) Page 5, line 126-127: The authors write that “any coding change is likely deleterious.” This is likely overly simplistic. A gene can be under purifying selection, however, it is not required that mutations at every site would be deleterious. This sentence should be revised.

We have revised this sentence following the referee’s suggestions (lines 136-138): “In genes under strong purifying selection the majority of nonsynonymous changes are likely mildly deleterious, hence an increased accumulation of deleterious nonsynonymous mutations results in a higher dN/dS ratio⁴¹.”

3) Top of page 7: The authors write that “reduced purifying selection is expected to translate into coding mutations with strong deleterious effects”. The way that this sentence is worded is technically incorrect. Reduced purifying selection does not result in coding mutations. Assuming that the mutation rate is not changed (which appears to be the assumption throughout this study), then the rate of appearance of deleterious mutations is the same. Reduced purifying selection affects what happens to those deleterious mutations in the population. Reduced purifying selection can increase the fixation probability of the deleterious mutations, allowing them to accumulate as substitutions. Also, I don’t think the authors can conclude that the mutations (or substitutions) would have “strong deleterious effects”. Relaxed purifying selection is more likely to allow weakly deleterious mutations to persist. Strongly deleterious ones will be purged out, regardless of the demography of the population. This sentence should be revised in light of my comments.

We agree with the referee’s criticism and have changed the sentence accordingly (lines 168-171): “In addition to elevated rates of nonsynonymous mutation accumulation, reduced purifying selection in sexual oribatid mite lineages is expected to result in fixation of nonsynonymous mutations that have stronger deleterious effects as compared to asexual lineages.”

REVIEWERS' COMMENTS:

Reviewer #1 (Remarks to the Author):

The authors have taken all my comments into account. They have added missing information (for example, by providing more details on the next-generation sequencing techniques, how individual genetic differences were taken into account in the transcriptome analyses and testing for positive selection, and many small comments). I also found it a very good idea that the authors added the BUSCO analyses. I find that the manuscript has improved a lot. In my opinion, it is now ready to be accepted for publication.

Reviewer #2 (Remarks to the Author):

Overall, the authors were very responsive to my previous comments. They have addressed the major technical concerns that I had on the previous version of the manuscript.

I have a few additional minor comments:

1) The authors write in the Introduction, "This benefit derives from the ability of sexual reproduction to decouple linked loci with different fitness effects generated by genetic drift, which increases the effectiveness of background selection to purge mildly deleterious mutations (HillRobertson effect) 8–11 ." This isn't strictly true and could be presented a lot better. I would suggest modifying this sentence to read, "This benefit derives from the ability of sexual reproduction to decouple linked loci with different fitness effects, which increases the effectiveness of purifying selection to purge mildly deleterious mutations and reduces Hill Robertson effects. 8–11"

2) Later, the authors write, "Population size influences the strength of genetic drift in a population, which makes it an important determinant for the speed of mutational decay through the Hill Robertson effect and Muller's ratchet in asexual lineages 13–15". Again, I think this sort of misses the point. The Hill Robertson effects aren't so important here. Rather, population size influences the strength of drift in the population which in turn influences what happens to weakly deleterious mutations in the population. In very large populations, purifying selection can out compete the effects of drift and weakly deleterious mutations will be eliminated from the population. In small populations, weakly deleterious mutations can sometimes drift up in frequency and become fixations. I would suggest citing Kimura 1963 on this topic (<https://www.ncbi.nlm.nih.gov/pmc/articles/PMC1210420/pdf/1303.pdf>) and revising this sentence.

3) The section heading, "Results on the transcriptome scale are supported by a larger taxon sampling" should be revised. Based on the results presented in both sections, I do not think the results are necessarily directly concordant. The transcriptome sampling suggests the sexual lineages have higher dN/dS ratios. The results in this section, however, do not

support any difference. Thus, they're not consistent with each other. This isn't a problem and I think the authors explanation is correct. But, I do think that the section heading should be fixed to better reflect the results.

4) I think it would be good in the Discussion to mention that this study conditions on those asexual lineages that made it and survived over long periods of time. Could previous studies considering younger asexual lineages by seeing accumulation of deleterious mutations in lineages that are destined to go extinct and have a meltdown prior to 1.1 MY in the future?

Reviewer #1 (Remarks to the Author):

The authors have taken all my comments into account. They have added missing information (for example, by providing more details on the next-generation sequencing techniques, how individual genetic differences were taken into account in the transcriptome analyses and testing for positive selection, and many small comments). I also found it a very good idea that the authors added the BUSCO analyses. I find that the manuscript has improved a lot. In my opinion, it is now ready to be accepted for publication.

We appreciate the very positive evaluation of our revised manuscript and like to again thank the reviewer for the helpful comments and suggestions made during the revision process.

Reviewer #2 (Remarks to the Author):

Overall, the authors were very responsive to my previous comments. They have addressed the major technical concerns that I had on the previous version of the manuscript.

We appreciate to hear that our revisions met the reviewer's approval and like to thank the reviewer for the comments and suggestions made that helped to improve the manuscript.

I have a few additional minor comments:

1) The authors write in the Introduction, “This benefit derives from the ability of sexual reproduction to decouple linked loci with different fitness effects generated by genetic drift, which increases the effectiveness of background selection to purge mildly deleterious mutations (Hill Robertson effect) 8–11 .” This isn't strictly true and could be presented a lot better. I would suggest modifying this sentence to read, “This benefit derives from the ability of sexual reproduction to decouple linked loci with different fitness effects, which increases the effectiveness of purifying selection to purge mildly deleterious mutations and reduces Hill Robertson effects. 8–11”

We agree that the way this sentence is worded is misleading. We have adopted the reviewers suggestion (lines 50-53): “This benefit derives from the ability of sexual reproduction to decouple linked loci with different fitness effects, which increases the effectiveness of purifying selection to purge mildly deleterious mutations and reduces Hill Robertson effects. 7-10”

2) Later, the authors write, “Population size influences the strength of genetic drift in a population, which makes it an important determinant for the speed of mutational decay through the Hill Robertson effect and Muller's ratchet in asexual lineages 13–15”. Again, I think this sort of misses the point. The Hill Robertson effects aren't so important here. Rather, population size influences the strength of drift in the population which in turn influences what happens to weakly deleterious mutations in the population. In very large populations, purifying selection can out compete the effects of drift and weakly deleterious mutations will be eliminated from the population. In small populations, weakly deleterious mutations can sometimes drift up in frequency and become fixations. I would suggest

citing Kimura 1963 on this topic

(<https://www.ncbi.nlm.nih.gov/pmc/articles/PMC1210420/pdf/1303.pdf>) and revising this sentence.

We have revised the sentence according to the reviewers suggestion and included the suggested reference (lines 60-63): “Population size influences the strength of genetic drift, which makes it an important determinant of mutational load and hence mutational decay of a population¹².”

3) The section heading, “Results on the transcriptome scale are supported by a larger taxon sampling” should be revised. Based on the results presented in both sections, I do not think the results are necessarily directly concordant. The transcriptome sampling suggests the sexual lineages have higher dN/dS ratios. The results in this section, however, do not support any difference. Thus, they’re not consistent with each other. This isn’t a problem and I think the authors explanation is correct. But, I do think that the section heading should be fixed to better reflect the results.

We agree with the reviewers concerns. To meet formatting requirements we restricted the section heading to < 60 letters and simplified it (line 179): “Analyses of the larger taxon sampling.”

4) I think it would be good in the Discussion to mention that this study conditions on those asexual lineages that made it and survived over long periods of time. Could previous studies considering younger asexual lineages by seeing accumulation of deleterious mutations in lineages that are destined to go extinct and have a meltdown prior to 1.1 MY in the future?

Throughout the manuscript, we stress that the asexuals we investigated here are very old (lines 25-28, 82-85). We clarified this again now in the discussion at lines 262-265: “In contrast to this established consensus, our study showed no evidence for accumulation of deleterious nonsynonymous and synonymous point mutations in oribatid mite lineages that survived without sex for several million years.” We cannot make any prediction about the timeframe for extinction of other young asexual lineages that were found to accumulate mutations, because, albeit deemed deleterious, there are no measurements of the strength of fitness effects of these mutations. Thus, the timeframe for potential extinction through mutation accumulation could be very variable und remains speculative.